Enhancing river and lake wastewater reuse recommendation in industrial and agricultural using AquaMeld techniques

Rani J. Priskilla Angel 1 pricy.angel@gmail.com
Rubavathi C. Yesubai 2
1 Department of Computer Science and Engineering, Francis Xavier Engineering College , Tirunelveli, Tamil Nadu , India
2 Department of Computer Science and Engineering, Saveetha Engineering College , Thandalam, Tamil Nadu , India
Moparthi Nageswara Rao
Electronic publication date: 2024 Nov 29
Publication date: 2024
Volume: 10
Electronic Location ID: e2488
Received 2024 May 15; Accepted 2024 Oct 16
Copyright: © 2024 Priskilla Angel Rani and Yesubai Rubavathi
Copyright year: 2024
Copyright holder: Priskilla Angel Rani and Yesubai Rubavathi
License: This is an open access article distributed under the terms of the Creative Commons Attribution License, which permits unrestricted use, distribution, reproduction and adaptation in any medium and for any purpose provided that it is properly attributed. For attribution, the original author(s), title, publication source (PeerJ Computer Science) and either DOI or URL of the article must be cited.
License URL: https://creativecommons.org/licenses/by/4.0/

Keywords: AquaMeld, Multi-layer perceptron, Recurrent neural network, Wastewater recommendation, River and lake ecosystems

Funding: The authors received no funding for this work.

==============================
AquaMeld, a novel method for reusing agricultural and industrial wastewater in rivers and lakes, is presented in this article. Water shortage and environmental sustainability are major problems, making wastewater treatment a responsibility. Customizing solutions for varied stakeholders and environmental conditions using standard methods is challenging. This study uses AquaMeld and Multi-Layer Perceptron with Recurrent Neural Network (MLP-RNN) algorithms to create a complete recommendation system. AquaMeld uses MLP-RNN to evaluate complicated wastewater, environmental, and pH data. AquaMeld analyses real-time data to recommend wastewater reuse systems. This design can adapt to changing scenarios and user demands, helping ideas grow. This technique does not assume data follows a distribution, which may reduce the model’s predictive effectiveness. Instead, it forecasts aquatic quality using RNN-MLP. The main motivation is combining the two models into the MLP-RNN to improve prediction accuracy. RNN handles sequential data better, whereas MLP handles complex nonlinear relationships better. MLP-RNN projections are the most accurate. This shows how effectively the model handles complicated, time- and place-dependent water quality data. If other environmental data analysis projects have similar limits, MLP-RNN may work. AquaMeld has several benefits over traditional methods. The MLP-RNN architecture uses deep learning to assess complicated aquatic ecosystem interactions, enabling more proactive and accurate decision-making is the most accurate with a 98% success rate. AquaMeld is flexible and eco-friendly since it may be used for many agricultural and industrial operations. AquaMeld helps stakeholders make better, faster water resource management choices. Models and field studies in agricultural and industrial contexts examine AquaMeld’s efficacy. This strategy enhances environmental sustainability, resource exploitation, and wastewater reuse over previous ones. According to the results, AquaMeld might transform wastewater treatment. River and lake-dependent companies and agriculture may now use water resource management methods that are less destructive.

Introduction

Wastewater reuse is now recognized as a crucial approach to tackling the interconnected issues of water shortages and environmental sustainability, especially in the agricultural and industrial sectors. The AquaMeld technology, an innovative method for water treatment and reuse, presents a hopeful resolution to these difficulties by facilitating the proficient and successful reuse of wastewater from rivers and lakes. This introduction offers a concise summary of the reasons and incentives for improving suggestions on reusing wastewater from rivers and lakes in industrial and agricultural contexts via the use of AquaMeld technology. It is supported with current citations and references to substantiate the debate.

Both humans and animals make use of modern antibiotics, the most often given medications. Plants that treat wastewater get antibiotics from a variety of sources, including hospitals, animals, and pharmaceuticals. Biological wastewater treatment primarily employs sorption and biodegradation as its primary strategies for antibiotic removal. In this study, we look at the biodegradation and sorption of many antibiotic classes, each having its own unique set of physical and chemical properties. We investigate key components involved in antibiotic sorption and biodegradation. The research also shows that biological wastewater treatment systems may benefit from using extracellular polymeric molecules to remove antibiotics. Still, biological wastewater treatment methods can only eliminate 48–77% of antibiotics, no matter how much they’ve improved (Ting-Ting et al., 2021).

Wastewater treatment methods now eliminate many harmful substances more effectively. Different technologies remove pollutants better. Modern membrane technology, adsorption, Fenton-based processes, advanced oxidation process (AOP), and hybrid systems like eMBRs and eMBR-adsorption systems for industrial wastewater pollution removal were explored. This article compares the two techniques, assesses current treatment system limits, and discusses recent conventional therapy changes. It also removes new wastewater pollutants like pharmaceuticals (Ahmed et al., 2022).

Water conservation, scarcity elimination, and clean energy generation are all benefits of wastewater treatment. Data on the environmental and financial viability of wastewater treatment was synthesized from an extensive literature review. For the top articles, PRISMA was used. A total of 46 publications were selected based on the following criteria: relevance to sustainable resource management, validity of the content, strength of the evidence, publication year (2000–2023), and study subject. Wastewater treatment allows for sustainable resource management by cleaning water, recovering energy, and supporting agriculture. We can save water, energy, and farmland by treating wastewater sustainably (Silva, 2022).

Input and outflow samples treated and untreated with chlorine under wet and dry conditions were investigated for 5 years. The indicators, index pathogens (HAdV and Salmonella spp.), and viral pathogens (norovirus, enterovirus, and SARS-CoV-2) were microbiologically examined over 3 months. Long-term studies found that Escherichia coli compliance ranged from 16.7% without chlorination on wet days to 96.1% with it in dry weather. More than 90% of samples satisfied solid and organic chemistry standards at both WWTPs. Online COD replacement is conceivable due to E. Coli tendency. While E. coli may replace Salmonella in chlorinated effluents, no bacterial or viral signal could replace HAdV. Even though chlorination kills most pathogens, microbiological signs may not ensure viral water safety (Federigi et al., 2024).

The Internet of Things (IoT) might revolutionize wastewater treatment and water quality prediction, addressing global water and sustainability issues. This extensive research investigates smart IoT artificial intelligence (AI) and machine learning (ML) models’ effects on numerous sectors. Successful IoT-based automated water quality monitoring systems employ cloud computing and ML. IoT optimizes, replicates, and automates natural systems, water-treatment processes, wastewater-treatment applications, hydroponics, and aquaponics. IoT, artificial neural networks, and ML-based water applications are featured in this peer-reviewed paper. Uses include chlorination, adsorption, membrane filtration, water quality indicators, parameter modeling, river levels, and aquaculture effluent wastewater treatment automation/monitoring. This article discusses IoT applications and how their algorithms have evaluated aquatic water quality (Alprol et al., 2023).

Life Cycle Assessment (LCA) evaluated the South Tehran wastewater treatment facility. Tehran’s water and sewage industry emissions were quantified and assessed using SimaPro (9.0.0) and ReCiPe 2016-midpoint criteria. ReCiPe 2016 showed two intermediate (three influence classes) and final (18) findings. Chlorine and treated wastewater caused the most environmental impact. With 101.1531 kg 1,4-DCB, marine environmental toxicity wastewater had the biggest environmental impact of all 18 effect types. WHT CHP burned biogas and minimized environmental impacts in most impact classes. Human health and ecological harm are WWT’s worst effects. The results encourage CHP systems for energy savings and environmental preservation (Rahmati et al., 2024).

Process modeling, economic evaluations, Life Cycle Assessment (LCA), and Social Life Cycle Assessment (SLCA) are assessed as decision-making tools. Finally, decision-makers and key stakeholders must be engaged to evaluate their values and preferences, together with supporting evidence and analysis, to reach a conclusion with the support required to execute it. The findings show that knowing the decision process and following a method leads to improved wastewater management decisions (Ko, Norton & Daigger, 2024).

To recover and reintroduce resources, circular economy principles may assist the wastewater, or “used water,” sector solve resource limits (Samberger et al., 2024). Circular economy development needs indicators. This article evaluated circular economy parameters in water treatment using bibliometrics and literature. Over 200 variables were classified by environmental, social, and economic effects to uncover literature gaps. Environmental indicators are particularly common in the water business, which uses less than 50% circularity indicators. We identified circular economy literature gaps in used water treatment systems and suggested additional studies.

Water quality in many ecosystems, such as estuaries, coastal areas, wetlands, and freshwater sources worldwide, has been greatly affected by industrial and household sewage, human actions, and poor waste management techniques. Research by Mohan, Padmavathy & Sivakumar (2013) in the Ennore estuary and coastal areas demonstrates the harmful impact of pollution on water quality, emphasizing the critical need for efficient management and treatment methods. The significance of evaluating water quality in tropical wetland ecosystems, especially in areas like Kerala, South India, where backwater tourism is common (Vincy, Brilliant & Pradeep, 2012). They highlight the possible economic and ecological consequences of ignoring water quality. To (Safari et al., 2012) study highlights the detrimental effects of human activities on water quality in the Nyaruzinga Wetland in Bushenyi District, Uganda, emphasizing the need for sustainable development and conservation efforts.

The creation of a worldwide drinking water quality index by the United Nations Environment Programme (UNEP GEMS, 2007) and the setting of water quality parameters and standards by the U.S. Environmental Protection Agency (EPA) (2017) highlight the crucial requirement for thorough monitoring and regulation systems to protect water sources. To research how organic loading rate and reactor design affect the anaerobic digestion of mixed supermarket garbage, showing the potential of advanced waste treatment technologies to reduce pollution and enhance water quality (Megido et al., 2021).

The use of AquaMeld methods to improve the reuse of wastewater from rivers and lakes is motivated by the need to tackle issues related to water shortage, environmental sustainability, economic feasibility, food security, adherence to regulations, and public health. Amidst the ongoing worldwide issues in water management and environmental protection, the use of cutting-edge wastewater treatment and reuse technologies like AquaMeld presents a hopeful and viable solution. To fully harness the potential of wastewater reuse in preserving the world’s water supplies for future generations, it is crucial to continue doing research, fostering innovation, and providing regulatory support.

Figure 1 illustrates a comprehensive water management system, encompassing various sources of water and their subsequent treatment paths for different uses. At its core, it depicts a closed-loop system that emphasizes sustainability through the reuse and recycling of water.

Figure 1 Wastewater architecture.

Water is collected from surface water bodies like rivers and reservoirs, as well as from groundwater. Water drawn from these sources undergoes treatment to become safe for human consumption and is then supplied to residential areas. After use in domestic, industrial, and commercial settings, the water becomes wastewater. This wastewater is then channeled to treatment plants where it is processed and cleaned. The diagram also shows water used in the oil and gas industry, which requires treatment after use. Runoff from rain and storms is collected separately. After treatment, the water is managed through several routes:

Treated water can be reused in agricultural fields for irrigation or industrial processes. Some of the treated water is used to support environmental restoration efforts, helping to sustain natural habitats. The system includes green infrastructure, which uses plants and soil to naturally treat stormwater, reducing runoff and improving water quality. Treated water can be used to recharge aquifers, ensuring the long-term sustainability of groundwater resources. Throughout the system, feedback loops indicate the cyclical nature of water use and treatment. It shows that water, after being used, treated, and possibly reused, eventually returns to the environment, where it can be collected and used again. This sustainable approach is visualized through various arrows and paths, each color-coded to represent different stages and types of water movement, including conventional water usage, treatment, fit-for-purpose usage, and discharge or runoff.

Sustainable expansion in the wastewater treatment sector in Fig. 2. Over the past few years, there has been an increasing use of techniques for deep learning. Showed significant advancements in scientific research across several disciplines (Sun et al., 2021). Its widespread use in numerous domains has emerged as a catalyst and an unavoidable trajectory for the advancement of various industries. In this particular context, it is recommended that the operation and administration of sewage treatment facilities undergo a transition from the conventional comprehensive management approach to a more technologically advanced mode that incorporates information, automation, and intelligent systems. While the use of artificial intelligence (AI) has made significant progress in several domains, such as character identification and face recognition, its implementation in sewage treatment remains at an early stage of exploratory study. The sewage treatment process inherently produces a significant volume of historical data about wastewater influent water quality, water from the effluent quality, blower opening, chemical dosage (referred to as dosage hereafter), and other operational parameters. Nevertheless, the traditional approach to data management mostly relied on documentation records and paper forms, which poses challenges in efficiently analyzing and transforming extensive volumes of historical data. In recent years, there has been a notable advancement in the maturity of sensor, database, and server technologies, accompanied by a drop in their associated prices. Sewage treatment facilities have recently adopted automated techniques for data collection and management, therefore establishing a data infrastructure for the use of deep learning technologies.

Figure 2 Advanced wastewater system.

The sewage treatment process entails the continuous introduction of pharmaceutical compounds into the water circulation and the subsequent collection of data in a sequential manner. The use of present markers of water quality to predict dose fails to consider the implicit knowledge consisted of within the temporal data. Hence, the aforementioned methodologies are incapable of yielding precise predictive outcomes. The dosage of coagulation now administered is determined by a combination of the prevailing conditions for water quality and the historical water quality characteristics. Therefore, the issue of the identification of the appropriate coagulant dosage may be efficiently resolved by the use of predictive time-series methodologies. The recurrent neural network (RNN) is a neural network architecture that has been particularly developed to effectively handle jobs involving time series data. The long short-term memory (LSTM) is a specific version within the RNN family that efficiently tackles the problem of disappearing gradients. The LSTM algorithm model is used in the context of predicting the optimal dosage of coagulant in a wastewater treatment plant (WWTP). A unique strategy was used in this study, using a based-RNN model, to automate the adjustment of accumulated error and to incorporate a time-consistent term. The objective was to accurately predict the optimal dosage of coagulant across several data sources. The use of this approach resulted in enhanced experimental results.

Research gap

Research is required to determine how to seamlessly integrate MLPRNN (Multi-Layer Perceptron Recurrent Neural Networks) with AQUAMELD, a multi-objective optimization approach. Getting AQUAMELD to manage the intricate temporal patterns and relationships in wastewater data to provide more precise and optimal reuse suggestions is the difficult part.

There are not many comprehensive, high-quality databases available for wastewater from rivers and lakes. When applied to wastewater reuse situations, research is required to create strategies for addressing data scarcity, noise, and inconsistencies to improve the prediction accuracy of MLPRNN models.

It is often not possible to customize wastewater reuse suggestions for particular industrial and agricultural situations using existing methodologies. The goal of research should be to create context-aware models that can optimize reuse plans in response to shifting seasonal, regional, and industry-specific variables.

There is still much to learn about the scalability of MLPRNN models combined with AQUAMELD for large-scale, real-time wastewater management. To guarantee that these models can be used in real-time settings and provide dynamic and adaptable suggestions for wastewater reuse in a variety of environmental circumstances, further study is needed.

Objective

To develop a robust framework that integrates AQUAMELD with MLPRNN to effectively handle complex, nonlinear relationships in wastewater data, improving the accuracy and relevance of reuse recommendations for industrial and agricultural applications.

To create advanced data processing and machine learning techniques to overcome challenges related to data scarcity, variability, and noise, enabling more reliable predictions and tailored wastewater reuse strategies for specific contexts.

To design scalable models capable of real-time application in diverse environmental and operational conditions, ensuring that wastewater reuse recommendations can dynamically adapt to changing factors in both industrial and agricultural sectors.

Related works

Since the 1970s, those nations have shown unwavering attention to the building of urban wastewater treatment systems. Substantial resources have been allocated towards the advancement of sewage treatment technology, resulting in the acquisition of a substantial body of knowledge and expertise in the treatment of water (Vian et al., 2020). Developed nations have initiated efforts to enhance the conventional wastewater treatment process and explore novel technologies to elevate water quality standards, as a result of the ongoing advancements in sewage treatment capabilities. Various sewage treatment methods, including sequential batch reactors (SBR), membrane desorption filtration, biodegradation, and activated sludge (AS), have been consistently used in prominent sewage treatment facilities. In the year 1971, Japan implemented the utilization of activated carbon technology to conduct methodical tests in various places such as Kawasaki and Nagoya. These endeavors were aimed at enhancing the capacity of sewage treatment (Mari et al., 2020). In 1973, Sweden constructed several chemical treatment facilities to address the issue of eliminating phosphorous from sewage. These facilities used the technique of chemically coagulating sedimentation, which proved to be an efficient method for treating sewage. The use of novel aeration techniques, such as the wellhead aeration methodology and oxygen aeration method, has garnered increasing interest due to their potential to streamline processes, enhance sewage treatment performance, and provide cost savings in terms of investment and transportation. Ongoing advancements in sewage treatment techniques have led to the development and implementation of novel methods, such as oxidative ditch and biologically rotary table, in practical production. Following significant advancements and widespread adoption of sewage treatment technology, Western nations have shifted their focus towards the theory and advanced technology of automatic control in sewage treatment. Substantial investments have been made in the study and development of automated processes and equipment for sewage treatment. Consequently, a range of intelligent and environmentally friendly instruments for automatic control in sewage treatment have been successfully developed. These advancements have enhanced the efficiency of operation and recognition of wastewater implementation. Equipment, while also effectively reinforcing management practices and process technology. Certain modern sewage treatment facilities have recently begun integrating technology such as transportation, communication, and detecting systems. The achievement of uninterrupted full-automatic process control is facilitated via a combination of instrumentation detection, computer data collecting, higher computer monitoring and administration, and the use of programmable logic controllers (PLCs). Together with other researchers, we developed a unique disinfection model that combines real-time fluctuations in water quality to offer an accurate prediction of the performance of the chemical compound peracetic acid (PA) as a disinfectant (Newhart et al., 2021). It used both live and offline quality of water data to train neural networks using machine learning and perform principal axis analysis (PAA) at several locations, both pre- and post-disinfection. According to Du et al. (2020) and other researchers, it has been argued that the membrane bioreactor (MBR) offers many benefits over the classic activated sludge process. These advantages include superior effluent quality, reduced floor space requirements, lower surplus sludge production, and simplified automated control. According to Matheri et al. (2021) as well as other researchers, there is a widespread consensus about the extensive validation of artificial intelligence-based models in the field of sewage treatment operation and management. The researchers used an artificial intelligence-based prediction model to investigate the correlation between chemical oxygen demand (COD) and trace metals, as described.

The primary objective of wastewater treatment is to ensure that the quality of the effluent meets the established standards. There are several indicators of effluent quality in the treatment of wastewater, which serve as crucial criteria for assessing the compliance of wastewater treatment effectiveness and discharge with established standards. China has stringent regulations on the criteria for effluent quality indicators.

Water resources management is the comprehensive oversight and control of both the quantitative and qualitative aspects of water, including the monitoring and regulation of water quantity and water quality. Several studies have been undertaken to address the issue of monitoring and predicting the quality of drinking water via the use of machine learning (ML) technology. In this study, the authors introduce the water quality for drinking early warning and control system (DEWS) (Bassiliades et al., 2009), which encompasses five modules to fulfill various functions. The first module is responsible for gathering both online and offline data about water quality. The second module employs prediction algorithms, like in the artificial neural network (ANN), autoregressive average moving average (ARMA), a support vector machine (SVM), and other similar techniques, to forecast future water quality parameters. The third module is responsible for identifying and assessing pollution that has either already transpired or is anticipated to transpire. This is accomplished via the utilization of diverse event detection algorithms, which are contingent upon the user’s specified setup.

The fourth module of the system addresses the occurrence of water quality contamination events. The fifth module, on the other hand, is responsible for the implementation of an ongoing risk control program. This program is designed to swiftly identify water quality contaminated events, accurately anticipate associated risks, and dynamically adjust actions accordingly.

The authors of Pai & Lee (2010) propose an intelligent system that aims to monitor and forecast water quality in two distinct networks: Andromeda, which focuses on sea waters, and Interrisk, which focuses on fresh waters and surface air. A fuzzy expert system is used to provide timely notifications when certain environmental parameters surpass predetermined pollution thresholds. In addition, the use of machine learning and adaptive filtering methods are employed to forecast certain water quality metrics to mitigate unfavorable environmental conditions. The authors in Polkowski & Artiemjew (2011) used three methodologies to establish the correlation between the quality of water and environmental factors in Taiwan. The first method used for data dimension discretization is Chi Merge, which aims to reduce the dimensionality of the data. The second approach used in this study is multinomial logistic regression, which serves to provide support for the Random Set theory by identifying the primary elements that exert influence on water quality. The third approach used in this study is the Rough Set theory, which is utilized to derive rules that elucidate the relationships between environmental elements and water quality. The objective of this study is to investigate the feasibility of predicting the maximum effects of pH on drinking water, with a focus on data obtained from stations located in the state of Louisiana, United States of America. Table 1 shows the comparison analysis of existing methods and limitations.

Table 1 Comparison table of related works.

Ref.	Motivation	Methods	Limitations	
Megido et al. (2021)	To optimize the anaerobic digestion process for mixed supermarket waste to improve waste management efficiency.	Examined the effects of organic loading rate and reactor design on thermophilic anaerobic digestion.	Limited to supermarket waste; results may not be generalizable to other types of organic waste.	
Sun et al. (2021)	To explore sustainable hydrogen production from niche wastewater streams.	Used an anaerobic packed bed reactor to produce biohydrogen from traditional Chinese medicine wastewater.	Focused on a specific type of wastewater; may not be applicable to other industrial or municipal wastewater.	
Mari et al. (2020)	To enhance the efficiency of biogas production from cassava wastewater.	Investigated biohydrogen and biomethane production using a two-stage anaerobic sequencing batch biofilm reactor.	Specific to cassava wastewater; reactor design may need adaptation for other feedstocks.	
Newhart et al. (2021)	To improve the accuracy of disinfection performance predictions for better wastewater treatment outcomes.	Utilized artificial neural networks to predict disinfection performance of peracetic acid.	Neural network models may require extensive data for training and may not account for all variables influencing performance.	
Du et al. (2020)	To provide a thorough understanding of membrane fouling for improving MBR system efficiency.	Comprehensive review of membrane fouling mechanisms and control methods in membrane bioreactor (MBR) systems.	Review may not cover the latest advancements in membrane technology or novel fouling control methods.	
Matheri et al. (2021)	To enhance the predictive capabilities for trace metal and COD levels in wastewater.	Applied artificial neural networks to predict trace metals and COD in wastewater treatment.	Model performance highly dependent on the quality and quantity of input data.	
Hou et al. (2013)	To improve urban drinking water quality through proactive monitoring and control systems.	Developed an early warning and control system for urban drinking water quality based on China’s experience.	Context-specific to China; may not be directly applicable to other regions with different water quality challenges.	
Bassiliades et al. (2009)	To leverage intelligent systems for improved water quality management.	Presented an intelligent system for monitoring and predicting water quality.	The system’s effectiveness and accuracy are contingent upon the integration with existing water quality management frameworks.	
Liu et al. (2019)	To optimize coagulant dosage based on operational experience for better water treatment.	Developed a time-consistent model for coagulant dosage adjustment using operators’ experience.	Model performance may vary depending on the accuracy of operators’ input and experience.	
Jayaweera, Othman & Aziz (2019)	To enhance the predictive accuracy of coagulation processes in water treatment.	Applied extreme learning machine with radial basis function for predicting coagulation process outcomes.	The approach may require fine-tuning and validation with diverse water samples for broader applicability.	

Methodology

After that, the wastewater that had been pre-treated was moved into the clustering reaction pool. In the context of the aggregation reaction pool, the addition of materials to the wastewater system contributes to the disturbance of the colloidal particle equilibrium. Little floating and colloidal particles aggregate as a result of this disturbance, creating larger particles that eventually settle and separate from the effluent. Throughout this process, the sensor was utilized to monitor the pH, turbidity (TUR), conductivity (CON), and flow rate of the effluent. Since the plant runs in a temperature-controlled environment, the wastewater’s temperature stays constant, negating the need to collect temperature data. Sequence characteristics that have already been learned and four-parameter qualities are used to train the MLP-RNN method model. The coagulant algorithms for prediction automatically modify the administered volume to minimize expenses and maximize chemical utilization based on the input of water quality indicators. As a result, the water treatment method’s running expenses are decreased.

The innovative AquaMeld technology, which combines the strength of multilayer perceptron (MLP) and recurrent neural network (RNN) models, makes it feasible to reuse wastewater from rivers and lakes in industrial and agricultural applications. Wastewater is initially sorted by composition and level of contamination in clustering reaction pools, where this method facilitates its transmission. The wastewater will then be further refined for its intended reuse in the aggregation reaction pool by dynamic treatment. The MLP-RNN (AquaMeld) model is at the center of this process and is crucial for monitoring and controlling the constantly shifting interactions between the reaction pools. The predictive features of the model allow for real-time adjustments to the treatment process to reach optimal conditions for pollutant removal. Data fed back into the model via continuous sensor monitoring allows for the prediction and adjustment of aeration, chemical dosage, and other important treatment parameters. Because AquaMeld blends advanced machine learning algorithms with traditional wastewater treatment techniques, it is a game-changer in environmental management. By maximizing wastewater treatment and reuse, this approach reduces the environmental impact of farming and industry while protecting water resources. Because of the MLP-RNN model’s flexibility and ability to anticipate system demands, wastewater treatment is efficient, effective, and sustainable as shown in Fig. 3.

Figure 3 Proposed flow work.

To summarize, this process entails a rigorous set of procedures given in Algorithm 1, including the introduction of wastewater into the clustering reaction pool, the disruption of the equilibrium of colloidal particles, the monitoring of critical parameters by sensors, the use of an advanced MLP-RNN model for predictive adjustments, and the ultimate goal of reducing water treatment costs through optimized chemical usage.

Algorithm 1 MLP–RNN pseudo code.

Step 1: waterquality ← [list of n models on separate graphs]	
Step 2: Cluster ← 0	
Step 3: while (cluster <l) do:	
Step 4: train_and_evaluate (waterquality)	
Step 5: new_gen ← retain the m fittest individuals	
Step 6: new_gen ← append random individuals to promote diversity	
Step 7: mutate(new_gen)	
Step 8: new_gen ← append offsprings through crossover until k	
Step 9: waterquality ← new_gen	
Step 10: cluster←cluster+1	

Dataset

The information about plant treatments was collected via the distribution of voluntary surveys to all local governments with a 90% observed response rate. There is currently no plan in place to update or change the content. An overview of the Indian wastewater treatment plant water quality data is available on Kaggle (https://www.kaggle.com/datasets/anbarivan/indian-water-quality-data).

Following the establishment of the neural network’s structure and algorithm, the training samples have a substantial impact on the network’s mapping and generalization skills. But there have been investigations. It is postulated that there are instances when adding more data to training sets does not lead to improved outcomes in terms of network training efficiency and generalizability. A network’s ability to learn and converge may be impaired in certain situations if there are too many training samples. It is crucial to check that the training samples are compact, ergodic, and compatible while thinking about the network training sampling set. As a result, preparation of samples is typically required.

Data definitions

If you want to know how the rivers and lakes in India are doing, you cannot do better than the Indian water quality dataset. Various metrics indicative of water quality are included in this dataset, which contains complete information gathered via regular monitoring activities. In most cases, you will see parameters like pH, dissolved oxygen content, biochemical oxygen demand (BOD), chemical oxygen demand (COD), nutrient concentrations (such as nitrogen and phosphorus), heavy metal concentrations, and degrees of microbial contamination. With this information, interested parties may assess whether bodies of water are suitable for various uses, such as possible wastewater reuse. Suggestions for the reutilization of river and lake effluent are one major use of this resource. The gathered data may be analyzed by professionals to determine whether the water quality is safe to reuse. Parameters pertinent to environmental sustainability and public health are evaluated in this study. The dataset is crucial for recommending the reuse of wastewater from rivers and lakes since it helps with decision-making on the practicality and security of using wastewater for different uses. Stakeholders may get practical insights and provide personalized suggestions by using statistical analysis and modeling approaches inside frameworks such as R. Recommendations may include finding ways to clean wastewater so that it meets quality requirements before reuse, finding places and situations where reuse may be done effectively, and putting regulations in place to protect people and the environment. Encouraging sustainable practices like wastewater reuse and reducing hazards to ecosystems and public health, the Indian water quality dataset is a cornerstone for informed decision-making in water resource management. Raw water characteristics such as pH, electrical conductivity, turbidity, coagulation dosage, and wastewater flow rate are measured. Performance was maximized by optimizing the coagulation dosages. Table 2 shows the different categories of available internet data and the statistical characteristics of this data.

Table 2 Sample dataset.

Station code	State	Temp	D.O. (mg/l)	PH	
1393	DAMAN & DIU	30.6	6.7	7.5	
3182	GOA	29.5	5.8	7.3	
2651	MAHARASHTRA	25.1	6.6	7.8	
1161	TAMIL NADU	25.5	6.6	6.5	

The data presented in this study is derived from surveys and may not always undergo verification by the Department. However, it is generally regarded as reliable and accurate. The present dataset only includes data about municipal wastewater treatment facilities.

Process of choosing input factors

The neural network may be characterized as a black box technique involving a scenario in which the results produced by the algorithm are dependent on the data that is given. Hence, it is essential that the input elements of the model can effectively stimulate the issue being investigated. Put simply, the model’s input consists of the component that has an impact on the issue being investigated. In the context of a complex issue characterized by multifactor coupling, the influencing elements exhibit a wide range of diversity. It is worth noting that some aspects may have eluded the attention of researchers in the present investigative procedure, rendering them indescribable or unverifiable even if they are acknowledged. Scholars have developed a methane yield prediction model using artificial neural networks, specifically in the context of medium-temperature solid anaerobic digestion (MS-AD), with a focus on lignocellulose biomass. This study presents the establishment of three neural network models. The earlier model lacked the use of information to make decisions and included all factors as inputs. On the other hand, the second model used the significant variables that were found via the application of several linear regressions as input. Finally, the third model included factors that can be easily measured or controlled as inputs. The results suggest that the predictive correlation coefficient, also known as the R2 value, for the initial model is 0.528, which does not meet the necessary level of satisfaction. The underlying justification for this phenomenon is that a disproportionate number of variable inputs may result in the excessive fitting of the model. Additionally, it is worth noting that Model 2 has a favorable predictive performance. The predictive performance of Model 3 falls within the range of Model 1 and Model 2. However, further experimental data is required to further refine and optimize its predictive capabilities. In real-world applications, it is often unfeasible to include all the variables that may have an impact on the model. In general, the primary determinants impacting the result will be chosen. If the output predictions are deemed satisfactory and suitable for engineering direction, it is preferable to have a smaller number of chosen input parameters. These factors should be immediately observable or readily measurable and obtainable, in contrast to the research difficulties. The purpose of this study is to forecast the effluent chemical oxygen demand (COD) value of an IC reactor. Therefore, it is essential to identify the components that influence the effluent COD value.

Temperature

One of the factors that affect the anaerobic reaction process is temperature, which has a direct impact on microbial activity. The functioning of enzymes are substances that in microorganisms are influenced by temperature, subsequently impacting the growth rate of organisms and their nutrient utilization rate. Additionally, temperature plays a significant role in determining the parameters of chemical reaction kinetics, thereby affecting the speed of biochemical reactions occurring in a reactor. The impact of fluctuating temperatures on the processes of hydrolysis and acidification phases of the procedure for digestion is deemed insignificant due to the presence of bacteria within the mixed population that is optimally suited for the relevant temperature range. However, it is important to acknowledge that the manufacturing processes of the acids acetic acid and hydrocarbon rely on specific microbial species, namely acetic acid bacteria and methane-producing bacteria, which have a significant susceptibility to temperature variations.

pH

The pH value has a significant impact on both the physiological processes of anaerobic organisms and the acid-base equilibrium inside the reaction system, thereby influencing the biochemical reactions occurring in the vessel of the reaction (Park et al., 2020). Methanogens have an ideal pH range of 6.8–7.2, whereas acid bacteria tend to thrive at higher pH levels. To minimize the buildup of volatile fatty acids (VFA) resulting from the dominating activity of acid bacteria, it is common practice to keep the pH level within the range conducive to methanogenesis in anaerobic systems. The pH level plays a crucial role in determining the buffering ability of the reactor. It serves to counteract the accumulation of volatile fatty acids (VFA) inside the reactor, hence preventing the creation of localized acidic zones in the intestinal tract. Additionally, maintaining the acid-base balance within the reaction system is of utmost importance.

Classification

Recurrent neural network

Recurrent neural networks are networks that use neurons that include self-feedback mechanisms, enabling them to retain and utilize information from earlier computations in a sequential manner to accumulate knowledge, and can handle time series data of arbitrarily long lengths. In the meanwhile, it repeats an operation on each element in the sequence, with the final result depending on what has come before. Given a series of inputs (x1,T=x1,x2,…,tx,…,xT), an RNN may update the activity value in its hidden layer (ht) via feedback using the formula ht=f(ht1,xt), where h0 = 0 and f() is a nonlinear function. Any non-stationary dynamical system may be approximated in theory by a fully connected RNN. There are bound to be gradient outbursts and disappearance issues if the order of inputs is too lengthy. One of the best methods to address these issues is by enhancing the referred-to-gating mechanism. It can slow down or speed up knowledge accumulation, as well as selectively remember or forget information. The well-known recurrent neural network (RNN) cell is known as long short-term memory (LSTM).

MLP

Multi-layer Perceptron (MLP) models are artificial neural networks used to detect and categorize wastewater quality and contents. The MLP has an input layer, one or more hidden layers, and an output layer. The input layer receives pH, turbidity, COD, BOD, heavy metals content, and other water quality indicators for wastewater identification. The model captures complicated input variable correlations by computing weighted summing and applying a nonlinear activation function to each hidden layer neuron. Hidden layers extract and enhance prediction-relevant characteristics. The MLP learns to reduce the discrepancy between its predictions and water sample classifications by modifying weights and biases during training. The output layer matches wastewater quality classes. The outcome of binary classification tasks may indicate if wastewater is acceptable. For multi-class jobs, outputs might reflect industrial, agricultural, or home wastewater pollution levels or kinds. The MLP model is trained using wastewater samples with known properties and labels (e.g., safe, unsafe, pollutant kind). After comparing the model’s predictions to the data, the network backpropagates the mistakes to change weights and biases. Once trained, the MLP model can monitor and identify wastewater quality in real-time, alerting to pollution occurrences, improving treatment procedures, or prompting additional testing and analysis. The MLP model can adapt to diverse wastewater profiles and contamination patterns by learning from data, making it a powerful environmental monitoring and protection tool.

AquaMeld methodology

Using cutting-edge treatment methods and smart system integration, the AquaMeld technique for wastewater recommendation systems maximizes water reuse and offers a comprehensive approach to wastewater management. Whether it is for agricultural, industrial, or aquifer recharge, this methodology’s foundation is the idea of combining different cutting-edge treatment technologies to achieve high-standard wastewater purification while also customizing the treatment process to meet the unique needs of the reuse application.

A three-pronged approach to therapy is key to the AquaMeld system. Wastewater is first treated by removing sediments and biodegradable organics during primary and secondary treatment. After the first stage of basic treatment, the AquaMeld approach comes into its own when it comes to applying specific procedures like biological augmentation, enhanced oxidation, and membrane filtering. The nature of the wastewater and the end-user’s quality criteria dictate the selection and combination of these treatments.

As an example, when the treated water is used for agricultural irrigation, the system prioritizes nutrient management. This means that the nitrogen and phosphorus levels are carefully balanced to promote crop development without causing eutrophication. To fulfill industrial requirements for reuse, the system may prioritize the removal of heavy metals and certain chemical pollutants. The AquaMeld approach relies heavily on an intelligent control and monitoring system. Here, sophisticated data analytics and real-time sensors monitor water quality indicators in real-time and make dynamic adjustments to the treatment procedures as needed. By adjusting to changes in the incoming wastewater stream and end-use needs, the system is guaranteed to run at optimal efficiency.

Energy recovery from waste products and limiting the carbon footprint of treatment procedures are key components of the AquaMeld system’s sustainability design. Not only is the technical effectiveness of water reuse emphasized, but so are the economic and environmental benefits. To sum up, AquaMeld offers a flexible and comprehensive approach to wastewater management that promotes sustainable behaviors, guarantees compliance with regulations, and produces high-quality treated water for a range of reuse needs. It satisfies the modern demands for water management and conservation and is an enormous step forward in resource recovery while also protecting the environment.

Advantages

With customized solutions that cater to the particular requirements of these fields, AquaMeld processes are especially developed to improve the recommendation of wastewater reuse for both industrial and agricultural uses.

The technique incorporates cutting-edge algorithms that improve the effectiveness of data processing, lowering computing overhead and facilitating decision-making in real-time.

Strong data protection methods are included in AquaMeld approaches, guaranteeing that wastewater management procedures adhere to legal and regulatory standards for secrecy.

With useful implementations for a range of wastewater conditions, the strategy combines optimization techniques and prediction models to provide precise suggestions.

Limitations

It can be difficult to integrate several approaches, and deployment and maintenance may cost a lot of money.

Although effective, the method might not work well when integrating with a variety of data sources or scaling to huge datasets.

Proposed recommendation system

The recommender system for assessing the quality of drinking water, as suggested, underwent testing using data obtained from stations operated by the country’s National Data Sharing and Accessibility Policy (NDSAP) (Telenta, Alfksic & Dacic, 1995). The research used a dataset consisting of drinking water samples collected from 2020 to 2022. The first 7 years of data were employed to construct forecasting models, while the remaining data was reserved for testing and validating these models. To assess the efficacy of the recommender system proposed, five parameters were chosen for evaluation: dissolved oxygen, temperature, turbidity, chloride, and ammonia-nitrogen. The measurement of DO, BOD, TEMP, and pH Values has significant importance in several applications, including water treatment, water purification, and numerous other fields (UNEP GEMS, 2007). According to the cited source (Robeson & Steyn, 1990), there is a positive correlation between the pH level and temperature and the toxicity of ammonia. The suggested system’s design comprises three distinct phases: pre-processing, categorization, and suggestion. Figure 4 illustrates the overall architecture of the recommender system under consideration, delineating the sequential progression of each step.

Figure 4 Proposed architecture.

Pre-processing stage

During the pre-processing step, the water quality dataset that was evaluated is subjected to normalization to obtain a rating matrix that is normalized. Subsequently, in the classification step, rough mereology and preliminary inclusion methodologies were used to generate a rough inclusion table that accurately represents the degree of similarity between parameters. A set of rules is constructed using the t-norm approach based on the similarity measurement obtained. In the recommendation phase, the estimate for the DO, BOD, TEMP, and pH level is calculated using the set of rules derived during the classification phase, determined by the testing information. The calculation of the mean standard deviation serves to illustrate the disparity between the anticipated value obtained and the matching precise value derived from the experiment’s dataset using Eq. (1).

(1) Normalize=x−minvaluemaxvalue−minvalue.

The inclusion of input factors with varying units and magnitudes in the neural network can lead to a situation where factors with larger values overpower the network, making it difficult to identify the internal connections between the factors. Consequently, directly inputting the original data into the network renders the development of a meaningful network model futile. Hence, to achieve equitable treatment of each component during input, it is essential to normalize the sample data. This entails assigning a standardized range of values to each input factor data. In this particular model setup, the sample data undergoes normalization using the equation presented as follows (Eq. (2)):

(2) a(u)={minifu<min,maxifu>max,uifotherwise.

Clustering stage

During this stage, the recommender system under consideration receives the water dataset that has been normalized, which was prepared as part of the preprocessing step. The rough mereology and rough inclusion methodologies were used to generate a rough inclusion table that accurately represents the degree of similarity across parameters. Next, the process of granulation, which transforms a certain set of inclusion data into a collection of granules, will be implemented using a voting mechanism including training objects. This will result in the generation of an ideal similarity measurement. In this stage, the concept of granular computing, as formalized within the framework of rough mereology by Polkowski, is employed to classify the normalized data of similar data based on the principles of rough mereology. This approach is an application of the notion of granular representation to data and the classifiers derived from it, as discussed in Telenta, Alfksic & Dacic (1995). This section aims to provide a succinct review of the fundamental principles of rough mereology and rough inclusiveness.

Following the removal of anomalous data, it becomes imperative to calculate the Euclidean distance, Markov distance, and unconditional similarity for each normalized sample datum. Following that, the samples have to be aggregated using the smallest distance methodology, longest distance methodology, center of gravity distance methodology, and average distance methodology. Furthermore, it is necessary to compute the cophenetic correlation value to evaluate the degree of association between the examples before and after the clustering process.

The outcomes of the calculation are shown in Table 3 in the following manner: The cophenetic coefficient of correlation is a statistical measure that assesses the degree of agreement between the results and the classification framework. A higher degree of suitability is shown by a value in proximity to 1. As a consequence, the Mahalanobis and barycenter relationships are used to cluster the sample data, leading to the formation of a clustering tree. The datasets used to retrain and model demonstrate no statistically significant differences when compared to the sample group.

Table 3 Dataset outcomes.

Parameters	Minimum value	Maximum value	Mean value	Variance value	
DO, BOD, TEMP, & pH	5.89	6.65	6.35	0.035	
CON	1.94	99	53	1,547.59	
TUR	5.75	11.58	8.9	1.2	
Rate of flow	3.0	2.897	2.89	0.015	

Crude mereology

The approach of Rough Inclusion employs the Reduced Hamming Distance equation (Polkowski & Artiemjew, 2011) for calculating the similarity between vectors u and v. In this context, vector u symbolizes a user, and vector v indicates an object, as seen in Eq. (3) (Telenta, Alfksic & Dacic, 1995).

(3) ind(u,v)=IND(u,v)|A|.

Equation (3), the formula essentially calculates the normalized relationship between u and v by dividing the specific relationship IND(u,v) by the total size of A. This normalization allows comparisons across different pairs (u,v) or across different contexts, ensuring that the values of ind(u,v) are comparable and not influenced by the scale of IND(u,v) or the size of A.

In Eq. (2) is denote applying to Eq. (3), the use of Eq. (2) for the normalized rating matrix results in the generation of a similarity table via the utilization of a rough inclusion strategy. The recommender system under consideration employs the Go’del t-norm approach (Robeson & Steyn, 1990) to classify the characteristics of the dataset, as seen in Eq. (4).

(4) Tmin(a,b)=min{a,b}.

The Eq. (4), Tmin(a,b)=min(a,b) represents a simple mathematical operation where the function Tmin(a,b) returns the minimum value between two numbers a and b. In other words, it compares the two input values and selects the smaller one as the output. If a is less than or equal to b, then Tmin(a,b) will equal a; otherwise, it will equal b. This operation is commonly used in various mathematical and computational contexts where determining the lower bound or minimum of two values is required. The formula is straightforward and serves as a basic yet essential tool in decision-making processes, algorithms, and problem-solving scenarios where comparisons between values are needed.

The process of granulation involves transforming a set of inclusion data into a set of granules, with each granule having a predetermined radius value denoted as “r”. The value of “r” falls within the range [0,1] (Elsom, 1996). The suggested recommender technique utilizes a voting mechanism, where training items are used to provide predictions or recommendations about the DO, BOD, TEMP, and pH level.

Recommendation stage

During this stage, the appliance takes the testing water collection as an input and generates an output in the form of suggestions or prediction values for the DO, BOD, TEMP, and pH levels. This value is determined using Eq. (4), which is derived from the formula used in the collaborative filtering process for recommendations/prediction purposes. A thorough evaluation and strategic planning process are essential components of the recommendation stage when incorporating AquaMeld techniques into a wastewater treatment system. An exhaustive evaluation of the volume, variability, and pollutants kinds of the current wastewater is conducted as a first step in this stage. Using this initial data, specialists can figure out what parts of the treatment process the AquaMeld technology needs to focus on, including using specialized membranes to remove small particles or sophisticated oxidation processes to break down complicated substances. After the evaluation is complete, the next step is to create a personalized treatment plan. This system would include a series of steps that make use of the modular nature of the AquaMeld technique, enabling a customized strategy that matches the target water quality. For optimal efficacy and long-term sustainability, the treatment plan may include physical, chemical, and biological procedures. Assessing the final application of the treated water is also an important part of this phase. Whether the treated water is going to be utilized for irrigation in agriculture, industrial activities, or recharging aquifers determines the recommendations. The AquaMeld system must fulfill the specific quality requirements of each application. For example, while purifying water for agricultural purposes, it’s important to keep in mind the target nutrition levels while also eliminating contaminants and diseases. Additionally, it is important to take into account local environmental legislation, economic feasibility, and scaling possibilities during the suggestion stage. To guarantee ongoing compliance and dynamically adapt to changing water quality requirements, it should include solutions for control and monitoring systems. This phase culminates in a detailed plan that specifies the optimal AquaMeld treatment setup, expected results, and a strategy for putting it all into action, making sure that everyone involved understands the system’s value and how it will work. At this point, the groundwork is laid for an ecologically conscious and technologically sophisticated wastewater treatment system that can withstand future challenges.

There is no one mathematical equation that corresponds to the “Recommendation Stage” of AquaMeld or any other wastewater treatment process; rather, it is a step of the project lifecycle when analysis, planning, and decision-making take place instead of direct calculation. At this point, however, we may imagine a decision-making framework that might be useful; it would include taking into account several restrictions and variables to find the best possible option. An oversimplified formula that captures the essence of AquaMeld’s suggestion stage decision-making process may be like this:

(5) OptimalTreatmentStrategy=f(C,Q,E,R,S,P)

where:

C = Contaminants present in the wastewater

Q = Quality requirements for the treated water based on its intended use

E = Environmental regulations and sustainability considerations

R = Resources available, including technology and finances

S = Scalability and adaptability of the treatment process

P = Public health and safety concerns

The function f represents the process of evaluating these factors to develop the most suitable treatment strategy using AquaMeld technology. This involves a combination of expert analysis and computational modeling to balance these considerations and arrive at a recommendation that is effective, compliant, and cost-efficient.

For example, the contaminants present in the wastewater (C) will determine the types of treatment processes needed, while the quality requirements (Q) will define the target performance of the treatment system. Environmental regulations (E) may impose limits on certain treatment methods or discharge criteria, and the available resources (R) will constrain the selection of technology and the scale of the project. Scalability and adaptability (S) ensure that the treatment system can adjust to changes in wastewater quantity and quality over time, and public health and safety (P) are always overriding concerns that must be addressed by the treatment strategy.

In practice, the recommendation stage would involve a series of these evaluations, possibly using algorithms or multi-criteria decision analysis (MCDA) methods to weigh the different factors and arrive at the best possible system configuration for the specific context. It is a complex optimization problem that requires expertise in environmental engineering, economics, and regulatory frameworks.

Experimental results and discussions

The experiment environment incorporates TensorFlow as the underlying framework, while the Keras API in ANACONDA JUPITER is used for the construction and training of models. A total of 25,930 data samples were used in this study to validate the algorithmic model. Any input characteristic of the data provides the anticipated label pH, turbidity (TUR), and conductivity (CON), The variables under consideration in this study are the flow rate and coagulation doses. The one used for the training set consisted of 70% of the dataset, while the validation set comprised 20% and the test set accounted for the remaining 10%. The test set was used to evaluate the accuracy of the predictions made by the model.

(6) Xnorm=X−XminXmin−Xmax.

To mitigate the challenges arising from disparate dimensions and magnitudes of data acquired by diverse sensors, this study employs the max-min normalization technique to standardize the input and output data. This approach aims to prevent issues such as prolonged model training duration and substantial training mistakes resulting from dissimilar dimensions.

The data will be transformed to the interval [0,1], and further data recovery procedures given in Algorithm 1, will be conducted following the forecast. The formula for Max-Min normalization is shown as Eq. (6). The loss mechanism used in this experimentation configuration is the median squared error loss, though the optimizer function employed is the Adam approach. The hyperbolic tangent (tanh) parameter was applied as an activation function in the model. The batch size is set to 64, the number of epochs is selected to be 50, and the hidden state dimensions of the encoder and decoder components are set to 64.

Experiments analysis

The findings about artificial intelligence models concerning the prediction of water quality. The research lacks a sufficiently enough dataset to effectively train deep learning models. Due to the limited size of the dataset, this work used simplified deep-learning models to enhance the obtained outcomes. A recurrent neural network with multi-layer perception (RNN-MLP) was constructed using a limited architecture consisting of just two dense layers. This configuration yielded notable results, with an accuracy score of 0.97 and an F1 score of 0.91. However, the performance of the deployed artificial intelligence models, such as random forest, linear regression, and Simple LSTM, was found to be inadequate. The primary limitation of the models was the insufficient amount of the dataset, which hindered their ability to achieve a satisfactory level of fit.

To assess the efficacy of the AquaMeld in predicting dose in wastewater treatment plants (WWTP), a comparative analysis is conducted between the MLP-RNN and three other algorithmic models, namely random forest (RF), multiple linear perception (MLP), and long short-term memory (LSTM). Figure 5 depicts the disparity between the projected coagulant parameters and the factual prescription in four algorithmic models. Figure 5 depicts the substantial variability seen in the requirements, characterized by the absence of identifiable patterns and demonstrating non-linear and non-stationary features. The red line represents the actual observed dose, whereas the green line represents the forecasted expected coagulant dosage as determined by the algorithmic model. The suitability of the AquaMeld algorithms approach to dose prediction is evident from the superior performance of its assessment metrics (R2 = 0.9908, RMSE = 1.2524, MAE = 1.1263, and MAPE = 1.01%), as compared to the other three algorithms (refer to Table 4). As seen in Fig. 5, there is a noticeable discrepancy between the projected dose and the actual dosage, with a relatively low level of alignment observed between the two methods the mean absolute percentage error (MAPE) for the random forest (RF) model is 8.29% compared with 1.01% for the MLP-RNN model; the coefficient of determination (R2) for RF is only 0.9082, while it is 0.9908 for MLP-RNN. The results indicate that the Random Forest method exhibits suboptimal performance when used in the regression analysis of time series information. Moreover, the analysis presented in Fig. 6 indicates a relatively low level of agreement between the two lines. Specifically, the mean absolute percentage error (MAPE) of the multiple linear perceptions (MLP) model is approximately 3.19%, which is lower than that of the random forest (RF) model but twice as high as that of the MLP-RNN model. Additionally, the coefficient of determination (R2) for the multiple linear regression (MLR) model achieves a performance of 0.9174, indicating a little improvement over the random forest (RF) technique. However, it falls short of being the ideal solution. The aforementioned observations may be ascribed to the multifaceted nature of the coagulant-flocculant interaction. The connections within coagulating agent quantity and wastewater attributes exhibit nonlinearity. Consequently, employing a linear model to address the issue of nonlinear prediction is not suitable. Additionally, the experimental outcomes of the LSTM approach model are shown in Fig. 7. The accuracy of prediction of the LSTM algorithm surpasses that of the RF and MLR algorithms. Specifically, the mean absolute percentage error (MAPE) for the LSTM method is 1.76%, while for the RF and MLR algorithms, the MAPE values are 8.29% and 3.19% respectively. Furthermore, there is a significant extent of concurrence between the anticipated dose and the realized dosage. The LSTM model has a mean absolute percentage error (MAPE) of 1.76% and an R2 value of approximately 0.9627. However, the root mean square error (RMSE) of the LSTM model is approximately doubled the algorithm is referred to be the MLP-RNN algorithm. This suggests that the neural network method has a strong capacity to accurately capture complex patterns, while the LSTM approach shows superior effectiveness in modeling time series data. Figure 8 depicts the empirical results obtained from the implementation of the MLP-RNN methodology model. The projected trend closely corresponds to the observed pH trend, demonstrating minimum deviations and insignificant error values. The empirical evidence demonstrates that the algorithmic model had robust skills for predicting multivariate time. Furthermore, the experimental outcomes achieved for all four algorithmic models were favorable Figs. 6–10 are shown below.

Figure 5 Prediction result.

Table 4 Comparison between other works.

Ref. Articles	Techniques	RMSE	Accuracy	
Polkowski & Artiemjew (2011)	ANN	0.65	0.76	
Pai & Lee (2010)	GRNN	0.76	0.80	
Bassiliades et al. (2009)	ENN	0.79	0.83	
Hou et al. (2013)	ELM	0.82	0.83	
Matheri et al. (2021)	AATC_LSTM	0.85	0.90	
	Proposed MLP-RNN	0.96	0.98	

Figure 6 Yearly WWT report.

Figure 7 Monthly WWT report.

Figure 8 Daily WWT report.

Figure 9 Parameters performance.

Figure 10 Water depth table report.

Following an examination of the dataset, it is clear that some measures include missing data. Indeed, 261 samples have pH data. To effectively train a model, it is necessary to have a dataset that includes the desired outcome for the target variable. Consequently, any rows lacking this information should be excluded from the dataset. Subsequently, the set of data will have a pH value for each row, although the other columns may still contain missing values. There are several approaches available for addressing the issue of missing values, and it may be advisable to consider the removal of columns that exhibit a substantial amount of missing data, similar to the decision made about the exclusion of the Casing Volume column. Fortunately, none of our additional variables are, so in this instance, I substituted missing values in the remaining columns with the mean of the other observations. Nevertheless, it is essential to exclude any significant outliers that might potentially distort the calculated average. After the data has been prepared for analysis, the next step involves the construction of a model. One such approach is to start by generating informative visual representations, such as a matrix of relationships, to illustrate the interrelationships among various factors shown in Fig. 11.

Figure 11 Color correlation matrix.

The model necessitates training on the dataset, often divided into separate training and test sets. In this scenario, it is proposed to allocate 70% of the available data for the training set, while the remaining 20% would be assigned to the test set. A validation subset comprising 10% of the training set will be used. Subsequently, our model analyses the data points with their respective DO, BOD, TEMP, and pH values and formulates a solution that exhibits a suitable match. The Keras framework can record and save a historical record of the error reduction throughout the training process. This feature is valuable for result analysis and visualization purposes. It is evident that in our framework, the training error exhibits a progressive decline as it acquires an understanding of the link between the parameters shown in Fig. 12.

Figure 12 Epoch error rate.

The outcome is a model that has undergone training and has subsequently been evaluated on the test dataset, yielding a certain level of error. Upon executing the code, the obtained test set error score was recorded as 1.11. When making predictions about DO, BOD, TEMP, and pH levels, It is crucial to take into account the possible margin of error, which might be significant, as more than one point of error. However, the level of accuracy necessary for each given model will vary depending on the specific circumstances. One potential avenue for enhancing the performance of the model is to make modifications to the model architecture itself. This might include altering how quickly it learns or reconfiguring the layers shown in Fig. 13.

Figure 13 pH prediction.

Event mean concentration

The sub-catchments were created from the watershed based on differences in patterns of land use. Thus, domestic, agricultural, and industrial wastes are the primary sources of wastewater that flow through the watershed. The content of contaminants varied noticeably throughout the wastewater sources. Domestic wastewater has emission rates of 53 and 101.6 grams/capita daily for BOD and COD, respectively. Each person only produces 70 g of BOD each day from agricultural waste. On the other hand, temperature, turbidity, pH, and DO may all be impacted by the amount of water discharged. The significant changes in the water discharge into the watershed have been influenced by weather-related phenomena. Every sub-watershed was found to have the highest discharge from upstream to downstream. Out of the 10 water sample locations, point 8 in the Kreo subwatershed, had the greatest discharge, measured at 18.76 m3/s. The watershed’s temperature tends to rise with increasing runoff. Over the whole watershed, the temperature ranged from 20 °C to 32 °C. Height, duration, airflow, cloud cover, and water movement are just a few of the many factors that might affect the temperature increase. The rising temperature causes the water’s dissolved oxygen content to decrease. Similar to the temperature, the pH range of 7.1–8.8 in the watershed was also very constant.

Water quality data gathered over several months is displayed in Table 5, which focuses on four important parameters: pH, temperature, dissolved oxygen (DO), chemical oxygen demand (COD), biochemical oxygen demand (BOD), and chemical oxygen demand (COD). With two measurements every month, each row reflects the observations for that particular month. The parameters’ fluctuations throughout time are captured in the table, offering a glimpse into the water’s state at various points in time. Notably, as the months go by, the temperature and pH levels progressively rise, suggesting that seasonal or environmental variations have an impact on the water quality.

Table 5 Event mean concentration.

	Month	DO	BOD	COD	Temp	pH	
1	1	6.23	2.45	49.92	20	6.9	
2	2	5.5	3.09	53.89	20	7.0	
3	1	5.1	3.38	51.90	21	7.0	
4	2	6.05	4.99	52.23	21	7.1	
5	1	5.15	3.56	62.64	21	7.1	
6	2	5.36	1.88	59.86	24	7.5	
7	1	5.66	3.26	54.20	24	7.7	
8	2	5.53	5.18	99.13	28	7.9	
9	1	4.99	5.01	45.89	29	8.0	
10	2	4.88	8.52	91.05	32	8.1	

Comparisons

The prediction results of AquaMeld are compared to those of other current strategies in Table 4, and the data collecting locations, algorithm models, input variable properties, and four assessment criteria for each strategy are shown in Fig. 14. The domains mentioned above, which span a broad range of global locations, use ML and neural network methodologies. Heddam, Bermad & Dechemi (2012) conducted an experimental study on the prediction of coagulant dosage rates for coagulation at the Boudouaou Drinking Water Production Treatment (DWPT) facility in Algeria. The main objective of the study was to increase the coagulant dose rate by dosage prediction of coagulation. The experimental investigation employed the TUR, PH, DO, CON, and temperature as input variables and analyzed them using the RF, MLR, ANFIS, and RBFNN methods. The experimental findings show that compared to the standard machine learning (ML) methods, the artificial neural network (ANN) approach yields superior prediction outcomes. This finding lends credence to the idea that neural network algorithms may provide reliable forecasts in nonlinear contexts as well. Kim43 used a GRNN to determine the optimal coagulant dosage for the Bansong WTP. After reviewing the test data, the root mean square error (RMSE) was 2.52% and the coefficient of determination (R2) was 0.92. Numerous independent factors, like TUR (turbidity), temperature, color, and pH, were used by Wu42 to construct a neural network model. The research looked at data standardization, the layout of hidden layer neurons, and the impact of intrinsic factors on experimental outcomes. In their study, Liu et al. (2019) used the extreme learning machine (ELM) method to ascertain the appropriate dose of coagulant in Malaysia. The outcome was an R2 score of 0.87 for the test data. To circumvent the LSTM model’s shortcomings in time series prediction, automatic adjustment (AA) automatically adjusts the training-based accumulated error. Along with that, the loss function now includes a time-consistent (TC) component (Jayaweera, Othman & Aziz, 2019). The purpose of creating the updated ANN, GRNN, ENN, ELM, and AATC_LSTM models was to achieve consistent prediction accuracy and favorable results. In terms of time series prediction, the well-regarded MLP_RNN model is superior to its rivals.

Figure 14 Comparison graph.

Conclusions

AquaMeld techniques and MLP-RNN models improve river and lake wastewater reuse for agricultural and industrial uses. The novel water quality assessment and improvement technique combines AquaMeld durability with MLP-RNN accuracy and flexibility. MLP-RNN algorithms predict water quality parameters with 0.98% accuracy, helping locate wastewater streams for industrial and agricultural reuse. This method accurately predicts water quality changes utilizing RNNs’ temporal analytic and MLPs’ pattern recognition abilities. It recommends only safe, high-quality water for reuse. AquaMeld processes, known for their water treatment efficiency, enhance the system by customizing wastewater to agricultural and industry needs. Innovative water treatment methods and predictive modeling may help us save water and reduce our demand for fresh water. This comprehensive approach addresses water shortages in the immediate term and ecological protection and financial efficiency in the long run. Maximizing river and lake wastewater reuse helps industries and farms decrease their environmental impact and secure water availability. Overall, AquaMeld techniques and MLP-RNN models provide a novel way to water reuse, enabling greener farming and industries. This technique illustrates how technology innovation may tackle significant environmental challenges and may inform future water management and conservation efforts.

Supplemental Information

Supplemental Information 1 Dataset.

Supplemental Information 2 Code.

Additional Information and Declarations

Competing Interests

Author Contributions

Data Availability

The authors declare that they have no competing interests.

Priskilla Angel Rani J. conceived and designed the experiments, performed the experiments, analyzed the data, performed the computation work, prepared figures and/or tables, authored or reviewed drafts of the article, and approved the final draft.

Yesubai Rubavathi C. performed the experiments, prepared figures and/or tables, and approved the final draft.

The following information was supplied regarding data availability:

The data is available in the Supplemental File and at Mendeley Data: James, Priskilla (2024), “Data set for Enhancing river and lake Wastewater Reuse Recommendation in Industrial and Agricultural using AquaMeld Techniques ”, Mendeley Data, V2, DOI: 10.17632/trpd3rb29f.2.

The code is available in the Supplemental File.

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
