# Peer review of "Enhancing river and lake wastewater reuse recommendation in industrial and agricultural using AquaMeld techniques"

_PeerJ Computer Science, doi:10.7717/peerj-cs.2488_

## Round 0.1 · original submission · Major Revisions

Dear Author(s)

Good work and a few suggestions from reviewers to make the article effective.
please complete and submit it

·

Basic reporting

The manuscript's current version is dangling and unclear, and it isn't easy to correlate and understand the abstract and the rest of the paper. The conclusion also needs to be in alignment with the formulated objectives and obtained results.

Experimental design

Not clear

Validity of the findings

It is not clear, needs proper discussion and comparison with existing work and critical writing to bring out the reason behind it.

Additional comments

Manuscript title: Enhancing River and lake wastewater Reuse Recommendation in Industrial and Agricultural using AquaMeld Techniques
Manuscript Number: 97980
Reviewers comments
A major revision is being suggested for further review and consideration. The manuscript's current version is dangling and unclear, and it isn't easy to correlate and understand the abstract and the rest of the paper. The conclusion also needs to be in alignment with the formulated objectives and obtained results. Following are the comments that need to be addressed for further review:
1. The abstract needs to be rewritten, the work's motivation is unclear, and the results must be included with proper quantitative support.
2. Introduction: this needs to be enhanced with the help of recent articles published in the domain. The identification of the research gap and formulation of objectives are very poor and unclear. Hence, it is suggested that similar recently published articles in the domain be referred to to understand how to write and identify research gaps and formulate objectives, respectively.

3. Proper section numbering needs to be done.
4. The methodology is not clear; authors are suggested to explain the same properly with the help of a flow chart. Also, proper sections and subsections with numbering make it easier to understand.
5. Why did the authors list limitations in the Methodology section?
6. Results and discussions: What is the significance of "Methodology, parameters, and setting up the experiment" here? While a dedicated section of methodology has already been given,
7. The results are incomplete and not clear; all the quantitative values must be presented in tabular form, and proper discussion and comparison with the existing published work are necessary to analyze the quality of the proposed method. There are multiple criteria for real-time implementation.
8. The conclusion is not in alignment with the title abstract and,….., all must be in alignment. Hence, it is suggested that major findings with respect to the formulated objectives be rewritten with quantitative support in 200 words (max).
I wish authors a great success.

·

Basic reporting

1. I suggest adding a detailed summary table of the related studies with appropriate
parameters.
2. The overall structure of this paper is good and well explained with proper explanations.
3. State of the art comparison is not sufficient.
4. Please add datasets details.

Experimental design

1. In what ways does the augmented snake optimization contribute to the efficiency and performance of the proposed model?

Validity of the findings

1.Compare your proposed method with these existing approaches. Highlight the advantages and limitations of each.
2.Provide a detailed description of the proposed technique. Explain howAquaMeld Techniques and contribute to data confidentiality and accuracy.
3.Reflect on the limitations of your proposed framework. Address any challenges faced during implementation.

·

Basic reporting

1. The contribution of this article is not highlighted in the abstract, and quantifiable indicators are not provided.
2. There is no specific explanation of the latest research in the first section.
3. The combination of RNN and MLP needs to be further elaborated, especially presented in the form of graphs.
4. Further explanation is needed for formulas 3 and 4.

Experimental design

1. The experimental results were not effectively compared with other advanced methods, such as AQPSO-SOFNN and GK-ARFNN.
2. The number of references needs to be increased, and their authority is insufficient.

Validity of the findings

1. The combination of RNN and MLP needs to be further elaborated, especially presented in the form of graphs.

---

## Round 0.2 · accepted · Accept

Dear Author

Good work. Continue your important work

·

Basic reporting

ok

Experimental design

ok

Validity of the findings

ok

Additional comments

ok